# Phonon thermal Hall as a lattice Aharonov-Bohm effect

## Kamran Behnia[1*]

**1** Laboratoire de Physique et d'Étude des Matériaux
(ESPCI - CNRS - Sorbonne Université)
Université Paris Sciences et Lettres, 75005 Paris, France

* kamran.behnia@espci.fr

## Abstract

In a growing list of insulators, experiments find a misalignment between the heat flux and the thermal gradient vectors induced by magnetic field. This phenomenon, known as the phonon thermal Hall effect, implies energy flow without entropy production along the orientation perpendicular to the temperature gradient. Experiments find that the thermal Hall angle is maximal at the temperature at which the longitudinal thermal conductivity peaks. At this temperature, $T_{max}$, Normal phonon-phonon collisions (which do not produce entropy) dominate Umklapp and boundary scattering events (which do). In the presence of a magnetic field, Born-Oppenheimer approximated molecular wave functions are known to acquire a geometric [Berry] phase. Its origin is the difference in the spatial distributions of the positive charge of the nuclei and the negative charge of the electron cloud. The magnetic flux is negligible for the point-like nuclei, but not for the electron cloud. Therefore, the molecular wave-function becomes a complex number in the presence of the magnetic field. In a crystal, this implies a complex amplitude for transverse acoustic phonons, modifying three-phonon interference patterns, and twisting the quasi-momentum of the outgoing phonon. The rough amplitude of the thermal Hall angle expected in this picture is set by the wavelength, $\lambda_{ph}$, and the crest displacement amplitude, $u_m$, of transverse acoustic phonons at $T_{max}$. The derived expression is surprisingly close to what has been experimentally found in black phosphorus, germanium and silicon.

## 1   Introduction

Discovered almost two decades ago [1], the phonon Hall effect has attracted an intense attention in recent years. Nowadays, there is a long list of insulators in which the effect has been experimentally observed [2–17]. There is also an ample literature of theoretical proposals [18–26]. This field of investigation overlaps with others exploring the chirality and the angular momentum of phonons as well as their coupling with magnetism [27, 28].

The recent observation of phonon thermal Hall effect in elemental and non-magnetic insulators (phosphorus [12], silicon [15] and germanium [15] ), however, indicates that this is a fairly common phenomenon [29]. There appears to be a fundamental flaw in our understanding of heat propagation in insulators. This is surprising, because *ab initio* calculations have achieved an account of the experimentally measured thermal conductivity in silicon and other insulators within percent accuracy [30]. This remarkable accomplishment of computational solid-state physics implies that the phonon spectrum and phonon-phonon interactions are well understood. The unexpected phonon Hall response indicates that a crucial detail is overlooked in our fundamental picture of heat conduction in insulators [31, 32].

This paper proposes to find this missing element by looking at the consequences of an alliance between anharmonicity and non-adiabaticity. The idea is that an account of the phonon thermal Hall effect can be found by going beyond the harmonic and the adiabatic approximations [intriguingly treated next to each other in the textbook by Aschcroft and Mermin] [33].

Anharmonicity refers to the fact that the interatomic restoring force is not simply proportional to the atomic displacement. This is not exotic. All common solids are anharmonic with third-order elastic constants. Adiabaticity here refers to the the separation of the electronic and the nuclear motions driven by a large mass discrepancy, and warranted by the Born-Oppenheimer approximation [34]. It is known to break down in the presence of a magnetic field [35, 36]. The molecular Aharonov Bohm-effect [37, 38] is a demonstration that the nuclear and the electronic wavefunctions are complex functions of space and time at zero magnetic field. In a finite magnetic, ordinary acoustic phonons acquire a geometric [Berry] phase [39–42]. I will argue that, by altering Normal collisions between phonons, this generates a thermal Hall signal of an amplitude much larger than what was proposed before [43].

Figure 1 shows two experimental features, reported in previous studies and motivating the present paper. Panel a displays the reported data [12,15] for three elemental insulators. Panel b is reproduced from ref. [44].

Fig. 1a shows the temperature dependence of the longitudinal, $\kappa_{ii}$, and the transverse (i.e. Hall), $\kappa_{ij}$, thermal conductivities of silicon, germanium and black phosphorus. $\kappa_{ii}$ and $\kappa_{ij}$ are both normalized to their peak value in order to make an observation: *The thermal Hall signal peaks where the longitudinal thermal conductivity peaks and it decreases faster on both sides of the peak.* This is broadly true of all other cases (See Figure 3 in ref. [12]). We will call this material-dependent temperature $T_{max}$ and focus on the three elemental solids, because their longitudinal thermal conductivity is understood in great detail [30]. The correlation between the longitudinal and the Hall response may help to demystify the latter and its temperature dependence.

Fig. 1b shows the amplitude of the maximum $\kappa_{ij}$ in several insulators divided by the

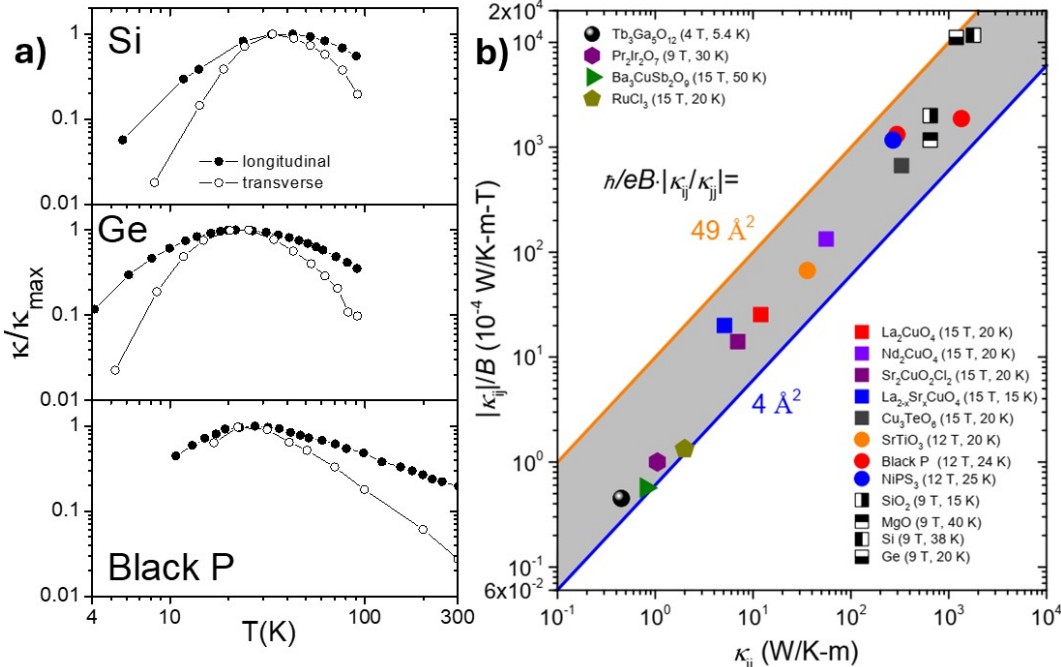

Figure 1: a) Transverse and longitudinal thermal conductivies in three elemental insulators normalized by their peak amplitude. The two conductivities peak at the same temperature and the transverse response decreases faster at both the warmer and the colder sides of the peak. b) The peak thermal Hall conductivity divided by magnetic field in a variety of insulators as a function of their maximum longitudinal thermal conductivity. While the latter varies by four orders of magnitude, the former remains proportional to it and their ratio displays little change.

magnetic field at which it was measured as a function of their maximum $\kappa_{ii}$. The sign of $\kappa_{ij}$ is neglected in this plot. One can see that $\kappa_{ii}$ changes by more than four orders of magnitude, which indicates that the phonon mean free path is very different in these insulators. The cleanest, as well as the simplest, are elemental (Si Ge and P). Their data symbols are located at the right side of the figure. The dirtiest (and the most complex) are at the left. remarkably, the peak amplitudes of $\kappa_{ij}$ and $\kappa_{ii}$ correlate with each other. In order words, *their ratio, the tangent of the Hall angle, does not exceed an upper bound.* Since the ratio remains small in laboratory magnetic fields ($\leq$ **0.01** in a magnetic field of $\sim$ 10 T), we will simply call the ratio the thermal Hall angle. Using fundamental constants, this ratio can be translated to a length scale. As seen in the figure, this length scale appears to be of the order of the interatomic distance (and does not correlate with the phonon mean free path, which is roughly proportional to the maximum $\kappa_{ii}$).

The aim of the present paper is to give an account of these experimental facts. Two questions are addressed: Why does the thermal Hall angle peak at $T_{max}$, (Fig. 1.a); and why its amplitude appear to be bounded (Fig. 1.b)?

The paper is organized as follows. Section 2 puts the notion of thermal Hall angle under scrutiny. A misalignment between the thermal gradient and the heat current density vectors implies a flow of heat without entropy production. As a consequence, the thermal Hall angle acts like the efficiency of a thermal machine, reminiscent of the Browninan ratchet introduced by Smoluchowski and popularized by Feynman [45]. Section 3 is devoted to anharmonicity. It argues that Normal phonon-phonon collisions become most prominent when $T \approx T_{max}$.

Section 4 is devoted to the fate of the Born-Oppenheimer approximation in the presence of magnetic field. The molecular Aharonov-Bohm effect was the first case of geometric (Berry ) phase [39, 40, 42]. In a magnetic field, atoms, molecules, and therefore phonons are expected to have a geometric phase. Section 5 considers how this phase can weigh on collisions between phonons and lead to a Hall signal. Finally, section 6 gives a rough estimation of the expected signal, which happens to be in good agreement with the experimental observation.

## 2   On the significance of a thermal Hall angle

Figure 2a is a sketch of a standard set-up for measuring the thermal Hall effect. The sample is sandwiched between a heater and a cold finger. This fixes the orientation of the heat density current, $\overrightarrow{J_q}$. Three thermometers (resistive chips or thermocouples) measure the local temperature at three different points of the sample. The difference between them leads to the quantification of the temperature gradient along $\overrightarrow{J_q}$ and perpendicular to it. The ratio of these two thermal gradients is no other than the tangent of the thermal Hall angle and identical to the ratio of off-diagonal to diagonal conductivity [or resistivity].

When the experiment finds a genuine difference between the local temperatures recorded by the two thermometers $T_1$ and $T_2$ of Figure 2a, there is a finite thermal Hall signal. It implies an angle between the two vectors $\overrightarrow{J_q}$ and $\overrightarrow{\nabla} T$ (Figure 2b) and therefore a process in which thermal energy flows without producing any entropy.

A temperature gradient implies entropy production. Non-equilibrium thermodynamics textbooks [46, 47] assume an 'entropy source', $\sigma^s$. It quantifies the rate of entropy production

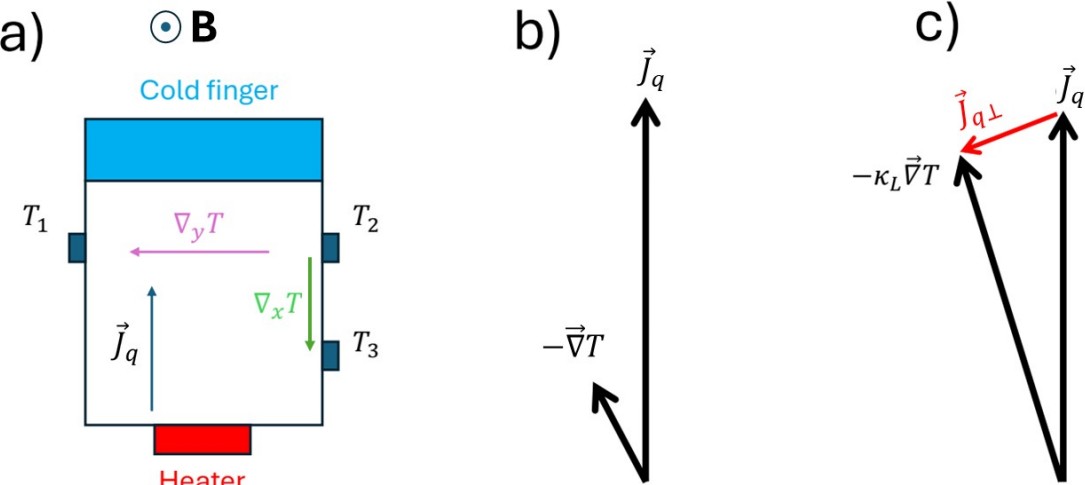

Figure 2: a) In a thermal Hall experiment, the sample is held between a heater and a cold finger. This allows to establish a finite heat current density vector, $\overrightarrow{J_q}$, along an orientation, which is often a crystallographic axis. Three temperature sensors allow measuring the temperature gradient along two orientations, which are parallel ($\nabla_x T$) and perpendicular ($\nabla_y T$) to $\overrightarrow{q}$. b) A finite $\nabla_y T$ means that there is a misalignment between the thermal gradient vector, $\overrightarrow{\nabla} T$ and $\overrightarrow{J_q}$. c) Such a misalignment implies a heat flow without entropy production in the direction perpendicular to $\overrightarrow{\nabla} T$.

per unit volume (units : W.K$^{-1}$m$^{-3}$). $\sigma^s$ and the entropy current density, $\overrightarrow{J_s} = \frac{\overrightarrow{J_q}}{T}$ are related with each other :

$$\sigma^s = \overrightarrow{\nabla} \cdot \overrightarrow{J_s} \tag{1}$$

The experimentally accessible quantity $\overrightarrow{J_q}$ is also linked to $\sigma^s$ [46, 47]:

$$\sigma^s = \overrightarrow{J_q} \cdot \overrightarrow{\nabla} \frac{1}{T} + \Sigma_\alpha J_\alpha X_\alpha . s \tag{2}$$

Here, $J_\alpha$ ($X_\alpha$) is an Onsager force (flux). The right hand side of this equation singles out a unique force-flux pair, namely. $\overrightarrow{\nabla} \cdot \frac{1}{T}$ (the thermal force) and $\overrightarrow{J_q}$ (the thermal flux).

Let us neglect thermodynamic forces and fluxes associated with flow of matter and charge (anything other than entropy). Then:

$$\sigma^s = \frac{1}{T^2} \overrightarrow{J_q} \cdot \overrightarrow{\nabla} T. \tag{3}$$

Equation 3 states that entropy production is the scalar product of the two vectors. When these two ($\overrightarrow{J_q}$ and $\overrightarrow{\nabla} T$) are not parallel to each other, a component of $\overrightarrow{J_q}$, which is perpendicular to $\overrightarrow{\nabla} T$, does not produce entropy (Figure 2c).

The efficiency of this engine can be defined as $\eta = J_{q\perp}/J_q$. It is easy to see that the tangent of the Hall angle, $\tan \Theta_H = \frac{\nabla_y T}{\nabla_x T}$, is linked to $\eta$, the sinus of the same angle, and therefore:

$$\tan \Theta_H = \frac{\eta}{\sqrt{1 - \eta^2}} \tag{4}$$

Experimentally, $\tan \Theta_H \ll 1$ and $\Theta_H \approx \eta$ is a good approximation, making the experimentally observed thermal Hall angle akin to an efficiency.

This situation is reminiscent of a Brownian ratchet [48, 49], introduced by Smoluchowski and popularized by Feynman in his famous lectures [45]. The original design consisted of a ratchet with a pawl and a spring (capable of rectification) coupled to a wheel which can rotate either clock-wise or anti-clockwise. However, the design can be simplified to any pairs of rectifier and non-rectifier Brownian 'particles' coupled to each other and kept at different temperatures [50]. Such an engine generates work with an efficiency below the Carnot efficiency [51].

There is a finite thermal Hall angle in an insulator when magnetic field becomes a rectifier inside a phonon gas subject to a temperature gradient. There is an obvious parenthood with the variety of stochastic ratchets combining thermal Noise and asymmetric potential [49]. Because of the fundamental differences between the phonon gas and a real gas, such an analogy has limits. The phonon number is not preserved and the group velocity of acoustic phonons does not depend on temperature. In contrast to a real gas, the flow of heat in a phonon gas is accompanied by a net flow of phonons. As we will see below, the magnetic field can become a 'ratchet' in this context.

## 3 Anharmonicity and phonon-phonon collisions

The harmonic approximation cannot explain the finite thermal expansion and the finite thermal resistivity of solids [33]. These are common features of real solids. Therefore, anharmonicity is widespread. Second-order elastic constants, defined by the linear relationships between strains and stresses, suffice for description of harmonic properties. On the other hand, higher-order elastic constants [52] are required to describe anharmonicity.

Discussing the origin of the negative thermal expansion of silicon between $\simeq 10$ K and $\simeq 120$ K [53], Kim and co-workers [54] distinguish between 'pure anharmonicity', associated with finite cubic and quartic elastic constants, and 'quasi-harmonicity', which assumes harmonic oscillators with volume-dependent frequencies. The latter can be captured by a finite Grüneisen parameter ($\gamma_i = \frac{V \partial \omega_i}{\omega_i \partial V}$) [33] and can be used to account for a thermal expansion of both signs. In contrast, an account of thermal resistivity requires invoking 'pure anharmonicity' and higher order terms in the interatomic potential. In this way, the energy of a phonon is altered by the presence of other phonons, irrespective of any change in the volume [54].

In their textbook [55], Hook and Hall give the following account of how phonons couple to each other. If strain modifies sound velocity, a traveling sound wave cannot remain indifferent to the presence of another sound wave. The coupling will show up in the phase modulation of the wavefront. Imagine a traveling phonon with a frequency of $\omega_1$ and a wave-vector of $\vec{q_1}$:

$$A = \Re[e^{i(\vec{q_1} \cdot \vec{r} - \omega_1 t)}] \tag{5}$$

Now, the presence of a phonon with a frequency of $\omega_2$ and wave-vector of $\vec{q_2}$ will modulate the phase of the first phonon [55]:

$$A = \Re[e^{i[\vec{q_1} \cdot \vec{r} - \omega_1 t + C \cos(\vec{q_2} \cdot \vec{r} - \omega_2 t)]}] \tag{6}$$

Hook and Hall introduced the dimensionless parameter $C$, which represents the strength of anharmonicity. Since $C \ll 1$, one can replace the exponential in equation 6 with its expansion :

$$A = \Re[e^{[i(\vec{q_1} \cdot \vec{r} - \omega_1 t)}[1 + iC \cos(\vec{q_2} \cdot \vec{r} - \omega_2 t) + ...]] \tag{7}$$

Neglecting the higher terms, the expansion becomes:

$$A \simeq \Re[e^{2i(\vec{q_1} \cdot \vec{r} - \omega_1 t)}$$
$$+ \frac{1}{2} C i e^{i[(\vec{q_1} + \vec{q_2}) \cdot \vec{r} - (\omega_1 + \omega_2)t]}$$
$$+ \frac{1}{2} C i e^{i[(\vec{q_1} - \vec{q_2}) \cdot \vec{r} - (\omega_1 - \omega_2)t]}] \tag{8}$$

The amplitude of this wave is real and can be expressed as:

$$A \simeq \cos(\vec{q_1} \cdot \vec{r} - \omega_1 t) -$$
$$\frac{1}{2} C[\sin[(\vec{q_1} + \vec{q_2}) \cdot \vec{r} - (\omega_1 + \omega_2)t] -$$
$$\sin[(\vec{q_1} - \vec{q_2}) \cdot \vec{r} - (\omega_1 - \omega_2)t]] \tag{9}$$

The first term of this equation represents the original wave with $\omega_1$ and $\vec{q_1}$ as frequency and wave-vector. The two other terms represent two new waves with identical amplitudes and and phase shifts ($\pi/2$ ). The first has a frequency and a wave-vector of:

$$\vec{q_3} = \vec{q_1} + \vec{q_2}; \omega_3 = \omega_1 + \omega_2 \tag{10}$$

The frequency and the wave-vector of the other one are:

$$\vec{q_3} = \vec{q_1} - \vec{q_2}; \omega_3 = \omega_1 - \omega_2 \tag{11}$$

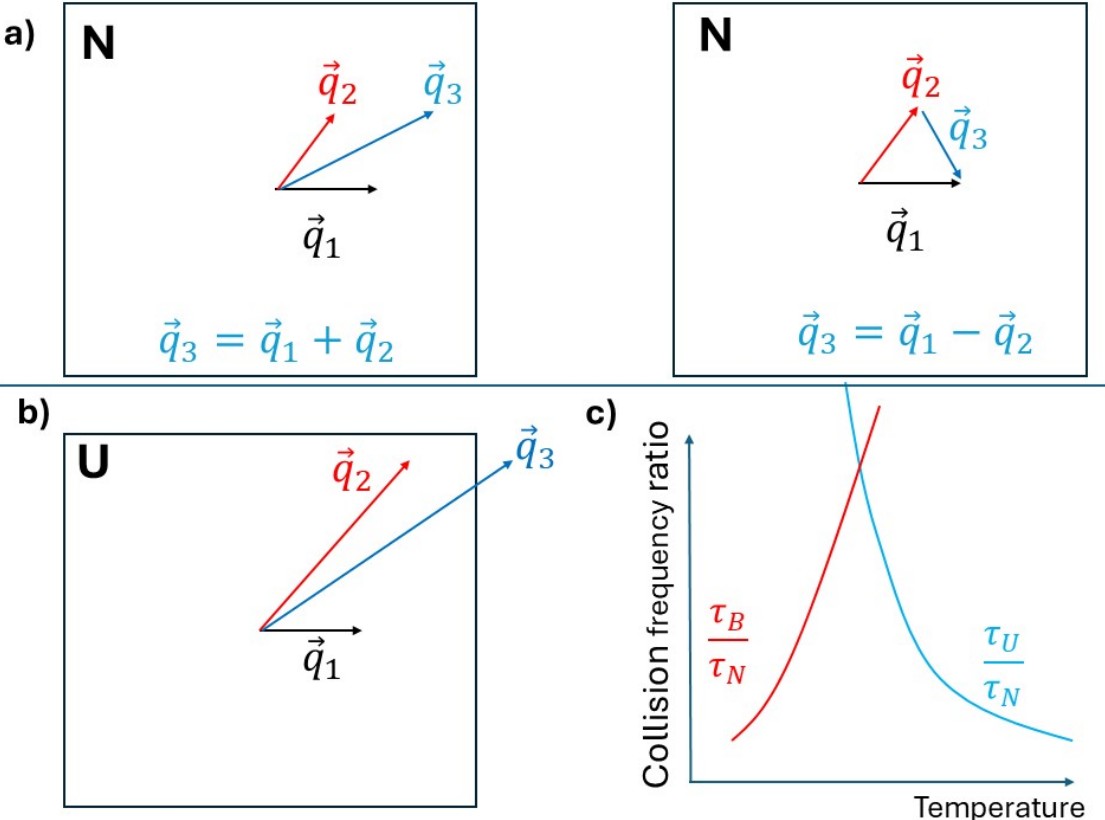

Figure 3: a) Two kinds of Normal three-phonon Normal collisions. A phonon with a $q_1$ wave-vector can absorb another phonon with a wave-vector $q_2$ (left) or emit a phonon with a wave-vector $q_2$ (right). These two Normal events can be transformed one to another by permuting the initial ($q_1$) and the final ($q_3$) phonons. They do not produce entropy. (b) When the sum of the wave-vector of the colliding phonons is sufficiently large, their collision is an Umklapp event, which generates entropy. Here the $q_3$ phonon is forbidden by the discrete symmetry of the crystal. (c) The relative frequency of Normal collisions respective to Umklapp and boundary scattering. At the peak temperature, the chance of a thermally excited phonon to suffer a Normal scattering event is the largest.

These equations represent three-phonon processes, in which a third phonon is created either by absorption or by emission. The four-phonon terms ($\mathcal{O}(C^2)$), known to matter in many crystals [56], will be neglected, for the sake of simplicity.

Hooke and Hall end their derivations of equations 10 and 11 by commenting that they are "more correctly regarded as a geometrical interference condition... than as a conservation law for momentum."

Absorption and emission events are distinct only in the presence of an arrow of time. If instead of beginning with $q_1$ and ending with $q_3$ one begins with the latter and ends with the former, an emission event becomes an absorption event and vice versa. Such events do not produce entropy. However, there are ph-ph scattering events, which do produce entropy (See Fig. 3b). If $\vec{q_3}$ gets out of the Brillouin zone, the scattering event is known as Umklapp. Since a lattice cannot host a sound wave with a wavelength shorter than twice the interatomic distance, $a$, it is impossible to have $|\vec{q_3}| > \frac{\pi}{a}$. When this is the case, a fraction of the quasi-momentum is absorbed by the crystal. Note that in this case, there is no way to start with the

forbidden final wave-vector.

The distinction between Normal and Umklapp events is not clearcut when $|\vec{q_3}| \approx \frac{\pi}{a}$ [55, 57]. However, despite this fuzziness, the statistical distinction keeps its conceptual utility for distinguishing between different heat transport regimes.

Phonon-phonon scattering is the main source of thermal resistivity at high temperature. At low temperatures, the wave-vector of thermally excited phonons is too short for Umklapp events. The major source of entropy production, absent large-scale disorder, is boundary scattering [31–33,55]. Experiments find that at low temperature (that is $T \ll T_{max}$), the thermal conductivity in clean crystals is set by sample size irrespective of the presence of point-like defects (chemical and isotopic impurities) [32]. The extracted phonon mean free path is of the order of the sample thickness and phonons travel ballistically .

Thus, in a perfect but finite crystal, there are two ways to produce entropy despite wave-like propagation. Either through boundary scattering or by generating a wavelength disallowed by the lattice discrete symmetry. These two routes for dissipation set the temperature dependence of the longitudinal thermal conductivity. It increases with cooling when the main driver of resistivity is Umklapp phonon-phonon scattering (which rarefies as the wave-vectors of colliding phonons shrink). It decreases with cooling when it is dominated by boundary scattering rate (due to the change in phonon population). The peak of thermal conductivity occurs when the two regimes, ballistic-extrinsic (low temperature and dominated by sample size) and the diffusive-intrinsic (high temperature and dominated by ph-ph collisions) meet.

This picture is to be complemented by considering the occurrence of Normal scattering events. As early as 1966, Guyer and Krumhansl [58] identified four distinct regimes of thermal transport regimes as a function of the hierarchy between different scattering times, including the one associated with Normal events. In addition to the ballistic and diffusive regimes discussed above, in the vicinity of the peak thermal conductivity. hydrodynamic regimes driven by Normal collisions between phonons have been postulated [58–60] and experimentally detected in a variety of insulators [61–67].

The time between two Normal events, $\tau_N$, and the time between two Umklapp events, $\tau_U$, decrease with warming. In both cases, there are simply more phonons and more collisions at high temperature. However, $\tau_U$ decreases faster because it includes an additional factor due to the change in the typical phonon wave-vector. Therefore $\tau_U/\tau_N$ decreases with warming. The time scale for boundary scattering is the ratio of sample size to sound velocity ( $\tau_B \simeq d/v_s$ ) and is roughly temperature-independent. Therefore, $\tau_B/\tau_N$ increases with warming (Fig.3c). Now, thermal conductivity peaks at the cross-over between the intrinsic and scattering regimes of scattering, where Normal scattering events, which do not produce entropy can weigh most (Fig.3b). Warming above temperature would reduce their frequency compared to Umklapp events. Cooling below this temperature would disfavor them in the face of competition by boundary scattering.

This simplified picture ignores the multiplicity of phonon modes with distinct velocities and the frequency dependence of $\tau_U$ and $\tau_N$. These subtleties notwithstanding, it is safe to assume the prominence of Normal phonon-phonon collisions near the peak. Indeed, hydrodynamic features have been observed near this peak by steady-state thermal transport experiments in many clean insulators [59,61–65]. The detection of these features require a specific hierarchy ($\tau_U > \tau_B > \tau_N$) of time scales.

The thermal Hall angle (which requires a flow of phonons without entropy production) peaks at the temperature at which Normal phonon-phonon scattering (which does not produce entropy) are most prominent. It is tempting to make a connection between the two. But how can the magnetic field influence collisions between phonons?

# 4 From the Born-Oppenheimer approximation to the geometric phase of phonons in a magnetic field

For almost a century, the Born-Oppenheimer (BO) approximation [34] has been a cornerstone of molecular physics and chemistry. Since the atomic mass, $M$ exceeds by far the electron mass $\mu$, the solution to the time-independent Schrödinger equation is expanded in orders of $\mu/M$ ratio. This allows disentangling the electronic and the nuclear motions. First, the electronic problem for fixed nuclei is solved. The higher orders of the expansion can then be used to quantify nuclear vibrations (phonons in a crystal) as well as their coupling to the electronic degrees of freedom [68–71]. In this approach, there is an Adiabatic Potential Energy Surface (APES) of electrons setting the nuclear trajectory.

The Born-Oppenheimer approximation is known to break down when the mixing of different electronic states cannot be ignored. The Jahn-Teller effect is known to arise in such cases [72].

Interestingly, one of the first identified manifestations of the Berry's phase [40] was "molecular Aharonov-Bohm effect" [38, 73], which occurs in the presence of a conical intersection between potential energy surfaces [74]. By making the electronic energy levels doubly degenerate, this can be pictured as a fictitious magnetic field.

Let us represent the nuclear coordinates by $R$. In the BO approximation, an energy eigenstate, $\Psi(R)$ is assumed to be the product of a nuclear wavefunction $\psi(R)$ and an electronic ket depending on $R$, $|\chi(R)\rangle$:

$$|\Psi(R)\rangle \approx |\chi(R)\rangle \psi(R) \tag{12}$$

Mead proposed distinguishing between a nuclear momentum operator, $\widehat{P} = -i\hbar\nabla_R$, the conjugate of the position operator for the total wavefunction, $|\Psi(R)\rangle$, and an effective momentum operator, $\widehat{\Pi}$, operating on $\psi(R)$. Their relationship can be written as [38]:

$$\widehat{\Pi}\psi(R) = \langle\chi(R)|\widehat{P}|\Psi(R)\rangle \tag{13}$$

Making the right hand side of the equation explicit leads to the following expression for nuclear momentum [38]:

$$\langle\chi(R)|(-i\hbar\nabla_R)|\Psi(R)\rangle = -i\hbar[\nabla_R\Psi(R) + \langle\chi(R)|\nabla_R\chi(R)\rangle] \tag{14}$$

The second term in the right hand side of this equation is recognizable as a Berry connection. If $\langle\chi(R)|\nabla_R\chi(R)\rangle$ is zero, there is no need to worry about the phase. However, when it becomes finite, the nuclei trajectory has a 'geometric vector potential'. In this case, the wavefunction cannot be treated as a real number [38]. Mead dubbed this effect 'the molecular Aharonov-Bohm effect' [37], referring to the celebrated proposal by Aharonov and Bohm in 1959 [75], in which the magnetic vector potential and the phase of the wavefunction play prominent roles.

As Resta has put it [40], the presence of a genuine magnetic field makes a qualitative difference. Now, in addition to a *possible* geometric phase, there is an *unavoidable* one, induced by magnetic field.

In 1988, Schmelcher, Cederbaum and Meyer [35, 36] examined the fate of the Born-Oppenheimer approximation in the presence of the magnetic field. They found that the simple extension of zero-field Born-Oppenheimer adiabatic approximation with no modification would make nuclei appear as "naked" charges, i.e., with no screening of the magnetic field by the electrons. This led them to to distinguish between the diagonal and off-diagonal terms of the non-adiabatic coupling and formulate what they called the 'screened Born-Oppenheimer approximation' [35]. In 1992, Mead [38] argued that this approach originally identified as

introducing "diagonal non-adiabatic correction" [35], and motivated by keeping gauge consistency, is equivalent to the addition of a geometric vector potential due to the finite magnetic field. The fate of this phase in molecules subject to strong magnetic field has been studied numerically in recent years [76, 77]. On the other hand, what happens in a crystalline lattice has beend barely explored (see [43] for an exception).

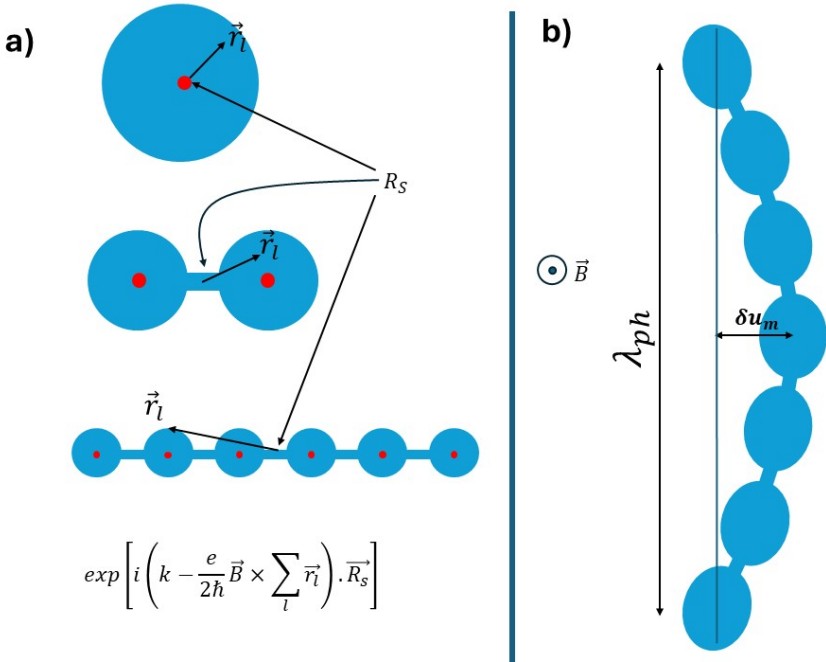

Figure 4: a) In the presence of magnetic field, an atom (top), a molecule (middle) and a chain of atoms (bottom) have a gauge-dependent phase. Expressed in the symmetric gauge, it is given by the vector product of the magnetic field and the coordinate of each electron with respect to the center of mass of the molecule. The ultimate origin of this phase is the fact that the magnetic flux swept by the electron cloud and by the point-like nuclei do not cancel out. b) A transverse acoustic phonon distorts the total amount of the magnetic flux swept by electrons. This distortion depends on two length scales, the wavelength $\lambda$ and the maximum atomic displacement, $\delta u_{max}$. These two length scales determine the change in the swept magnetic flux divided by the quantum of magnetic flux and set the phonon phase.

Schmelcher *et al.* [35], using the symmetric gauge, $\mathbf{A(r)} = \frac{1}{2}\mathbf{B} \times \mathbf{r}$, separate a phase factor of a homonuclear diatomic molecule correcting the pseudo-momentum, $\overrightarrow{k}$ of the molecular wave-function:

$$\exp[+i[\overrightarrow{k} - \frac{e}{2\hbar}(\overrightarrow{B} \times \sum_l \overrightarrow{r_l})] \cdot \overrightarrow{R_s}] \tag{15}$$

Here $\overrightarrow{r_l}$ represent the coordinates of the electron indexed $l$ with respect to the center of mass of the molecule. The latter's coordinates are represented by $\overrightarrow{R_s}$.

This expression invites several comments. Its first remarkable feature is that it does not depend on the ratio of the electron-to-nucleus mass ratio. Note also that this expression is gauge-dependent and it remains to be seen that it has any observable consequences. Moreover, as illustrated in Fig. 4a, such a gauge-dependent phase factor is present in an atomic wave-function, or in the wave-function of a chain of atoms. In each case, one needs to shift the

coordinate of the center-of-mass and set the amplitude of the pseudomementum as the sum of atomic pseudomomenta [35]. Finally, let us note that this phase factor of Equation 15 is the ratio of two areas: $(\sum_l r_l.R_S$ and $\ell_B^2$, with $\ell_B = \sqrt{\frac{\hbar}{eB}}$ being the magnetic length.)

The ultimate source of this phase factor is the difference in the spatial distributions of the positive charge of the nuclei and the negative charge of the electron cloud. The magnetic flux, which is gauge-invariant, is negligible for the point-like nuclei, but not for the electron cloud. The absence of compensation makes the molecular wave-function inherently complex in the presence of the magnetic field.

In order to see the consequences of this field-induced phase in a crystal, let us consider how a transverse acoustic phonon can change the magnetic flux swept by the electronic cloud of an insulating solid. The two relevant length scales are the phonon wavelength $\lambda_{ph}$, and the atomic displacement of $\delta u_m$ at the wave crest (Fig.4b)). Their product defines an area in which atoms are displaced from their equilibrium position. The geometric gauge-independent phase of such a phonon, indicated by our interpretation of equation 15, is this product divided by $\ell_B^2$. Equation 15 contains a sum of all electron coordinates too. Therefore, another relevant quantity is the total 'shuffled' amount of electric charge. Let us call it $Q = q_e e$, where $q_e$ is a real number and $e$ is the fundamental electric charge. This leads us to :

$$\delta\phi_B \approx q_e \frac{\lambda_{ph}\delta u_m}{\ell_B^2} \tag{16}$$

Let us now see how large $\delta\phi_B$ can be. When $B \simeq 10$ T and $T \sim 10K$, $\ell_B$ and $\lambda_{ph}$ are of the same order of magnitude (in the range of hundreds of angstroms) and $\delta u_m$ is two to three orders of magnitude smaller than both. As we will see in section 6, the dimensionless $q_e$ is of the order of unity. Therefore, in our range of interest the phonon phase is of the order of a few miliradians, comparable to the phase shift of an optical photon in a typical Kerr experiment on a magnetic material [78].

This small phase shift can affect the interference between two colliding phonons (Fig.4c). In the same magnetic field, two phonons with the same wave-vector but with different polarizations do not sweep an identical magnetic flux and therefore their $\delta\phi_B$ is different.

## 5 Normal phonon-phonon collisions in the presence of a magnetic field

Textbooks define phonons as quanta of lattice vibrations. The latter are waves of atomic displacements with an amplitude, which is a real vector. Complex waves have additional singularities [79] such as wavefront dislocations [80] where the wave vanishes because of its undefined phase. Let us now consider what can happen to sound waves acquiring becoming complex with a genuine phase in finite magnetic field.

Equation 6 can be revised by including $\delta\phi_B$, the phase difference between the original phonon and a second one, which by modifying the environment of the first, modulates its phase:

$$A = e^{i[\vec{q_1}\cdot\vec{r}-\omega_1 t + C\cos(\vec{q_2}\cdot\vec{r}-\omega_2 t + \delta\phi_B)]} \tag{17}$$

The Taylor expansion (Equation 8) becomes:

$$A \simeq e^{2i(\vec{q_1} \cdot \vec{r} - \omega_1 t)}$$

$$+ \frac{1}{2} C i e^{i[(\vec{q_1} + \vec{q_2}) \cdot \vec{r} - (\omega_1 + \omega_2)t + \delta\phi_B]}$$

$$+ \frac{1}{2} C i e^{i[(\vec{q_1} - \vec{q_2}) \cdot \vec{r} - (\omega_1 - \omega_2)t] - \delta\phi_B]} \quad (18)$$

Comparing equation 18 with equation 9, one can detect a difference. Now there is a $e^{2i\delta\phi_B}$ difference between the prefactors of absorption and emission events. The difference between the amplitude of the absorption and emission waves opens the road for a thermal Hall response. Let us call the angle between $\vec{q_1}$ and $\vec{q_2}$, $\theta$, and the component of $\vec{q_3}$, perpendicular to $\vec{q_1}$, $q_{3\perp}$. Then it is easy to see that for an absorption event, $q_{3\perp} = q_2 \sin\theta$ and for an emission event, $q_{3\perp} = -q_2 \sin\theta$. For both emission and absorption events, the final wave-vector ($\vec{q_3}$) becomes tilted with respect to the original wavevector ($\vec{q_1}$). Since the sign of the perpendicular wavevectors $q_{3\perp}$ is opposite for emission and absorption, the net tilt is zero when both types of events weigh equally.

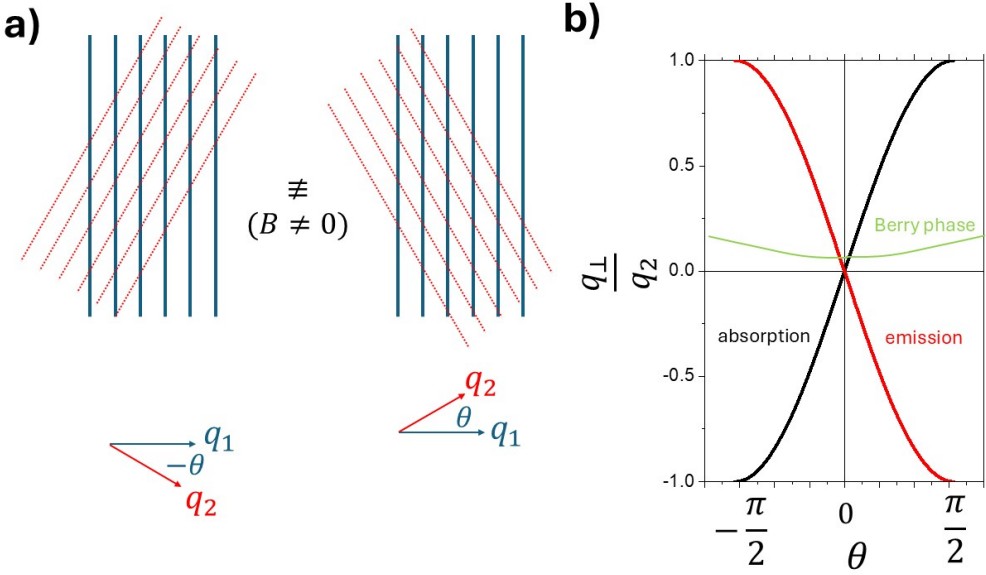

Figure 5: a) An illustration of interference between two lattice waves. In the absence of magnetic field, the two configurations are equivalent but not in its presence. b) The normalized amplitude of the perpendicular wave-vectors produced by absorption and emission events. In absence of a finite Berry phase, they would cancel each other, but not in its presence.

Consider a phonon with a $\vec{q_1}$ wave-vector traveling in an isotropic plane and encountering another phonon with a wavevector $\vec{q_2}$. In absence of a magnetic field, there is no difference if the angle between $\vec{q_1}$ and $\vec{q_2}$ is $\theta$ or $-\theta$. This would not happen in the presence of a finite $\delta\phi_B$. Even at $\theta = 0$, the latter can be finite. This is sufficient to generate a Hall signal (See Figure 5).

## 6   The rough amplitude

The simple picture drawn above can provide answers to two questions raised by experiments. Why does the thermal Hall angle peak at $T_{max}$? Because this is were the Normal phonon-

phonon scattering is most prominent. How can the magnetic field induce heat flow without entropy production? By influencing Normal phonon-phonon scattering, which is about wave interference, and not entropy production.

What about the amplitude of the thermal Hall signal?

A rigorous answer to this question is a task for *ab initio* studies starting from the phonon spectrum in each solid. In 2019, Saito and collaborators [43] proposed that in a nonmagnetic insulator with ordinary phonons, such as silicon, Berry phase can give rise to a detectable thermal Hall effect. However, their expected thermal hall conductivity was $10^{-6}WK^{-1}m^{-1}$. As seen in table 1, the experimentally observed thermal Hall conductivity in silicon [15] is 7 orders of magnitude larger! Anharmonicity, the central player in our scenario, was not mentioned there [43].

The amplitude of the thermal Hall conductivity correlates with the maximum longitudinal thermal conductivity. The latter is known to depend on the sample size [62]. The maximum thermal conductivity is not an intrinsic property. In silicon crystals [81] larger than the one in which the thermal Hall conductivity was measured [15], longitudinal thermal conductivity is several times larger.

In the picture drawn here, the Hall angle, the ratio of longitudinal to transverse thermal conductivities at $T_{max}$ (see table 1), is the central quantity to put under scrutiny. If the field-induced misalignment between the heat current and the temperature gradient is due to interference between phonons, then one expects it to be $\approx 2\delta\phi_B$.

| Solid | $T_D$(K) | $T_{max}$(K) | $\kappa_{ii}(WK^{-1}m^{-1})$ | $\kappa_{ij}(WK^{-1}m^{-1})$ | $\frac{\kappa_{ij}}{\kappa_{ii}}B^{-1}(10^{-4}T^{-1})$ |
|---|---|---|---|---|---|
| Si | 636 | 33 | 1880 | 10.6(9 T) | 6.3 |
| Ge | 363 | 20 | 610 | 2.5(9 T) | 4.6 |
| Black P | 306 | 24 | 1900-300 | 2 (12T) | 0.9-5.5 |

Table 1: Longitudinal and transverse thermal conductivity in silicon [15], germanium [15] and black phosphorous [12]. $T_D$ is the Debye temperature [62, 82] and $T_{max}$ the temperature at which both the longitudinal ($\kappa_{ii}$) and the transverse ($\kappa_{ij}$) thermal conductivity peak. The next two columns list their amplitudes at $T_{max}$. In black phosphorus, the two values for $\kappa_{ii}$ correspond to two different crystalline orientations [12, 62]. The last column represents the ratio of two conductivities, the Hall angle normalized by the magnetic field.

| Solid | Mass (a.u.) | a (Å) | $v_t$ (km/s) | $\lambda_{ph}(T_{max})$ (Å) | $\delta u_m(T_{max})$ (Å) | $q_e(T_{max})$ | $\delta\Phi_B(T_{max})$ $(10^{-4}T^{-1})$ |
|---|---|---|---|---|---|---|---|
| Si | 28.1 | 2.72 | 5.8 | 84 | 0.23 | 2.6 | 3.3 |
| Ge | 72.6 | 2.83 | 3.6 | 86 | 0.18 | 2.0 | 2.6 |
| Black P | 31 | 2.74 | 1.2-4.9 | 24-98 | 0.25 | 0.8-3.3 | 0.3-5.4 |

Table 2: Atomic masses and length scales of silicon, germanium and phosphorous. $a = (V_c/n_c)^{1/3}$ is the average distance per atoms (with $V_c$ and $n_c$ are the volume of the primitive cell and its number of atoms). $v_t$ is the transverse sound velocity used to extract the transverse phonon wavelength at T=$T_{max}$. The two $v_t$ values for Black P represent the lowest and the largest known [62]. $\delta u_m$ is the crest atomic displacement at T=$T_{max}$. The last column is the the rough amplitude of the phonon phase shift at $T_{max}$, to be compared with the last column of table 1.

Let us now quantify $\delta\phi_B$ in the three elemental insulators listed in table 1. Let us examine, $\lambda_{ph}$, $\delta u_m$ and $q_e$, the three quantities to be injected in equation 16.

The most familiar quantity is the wavelength of an acoustic phonon. Consider a linearly dispersive transverse phonon with a wavevector of $q_t = \frac{2\pi}{\lambda_t}$ and a velocity of $v_t = \omega_t/q_t$. Its wavevector is simply proportional to thermal energy and its wavelength proportional to the inverse of temperature. At, $T = T_{max}$, this wavelength becomes equal to :

$$\lambda_{ph}(T_{max} \approx \frac{h v_t}{k_B T_{max}} \tag{19}$$

Note that at T=0, $q_t$ vanishes and the wavelength of an acoustic phonon becomes infinite. This of course never happens in a solid of finite size. However, in realistic experimental set-ups, we are far from this limit. Given that sound velocity in all known solids is between 1 to 20 km/s, the wavelength of thermally excited phonons is about in the range of $\sim$ **10** nm at 1 K and $\sim$ **1**$\mu$ m at 10 mK. The samples studied in these experiments are much larger.

The amplitude of $\delta u_m$, the atomic displacement at the crest of a lattice wave can be estimated by its link to the energy of the thermally excited phonon mode: $M \delta u_m^2 \omega_t^2 = k_B T_{max}$, where $M$ is the atomic mass [83]. Combined with $\hbar \omega_t(T_{max}) = k_B T_{max}$, it becomes:

$$\delta u_m(T_{max}) \approx \frac{\hbar}{\sqrt{M k_B T_{max}}} \tag{20}$$

Thus, $\delta u_m$ proportional to the de Broglie thermal length of the vibrating atom. Both diverge at T=0. However, this divergence is even weaker than the zero-temperature divergence of $\lambda_{ph}(T)$. In our temperature range of interest, $\delta u_m$ remains shorter than the interatomic distance, which we call, $a$ (See table 2).

Among the three parameters of equation 16, estimating $q_e$ is the least straightforward. One one hand, an acoustic phonon, which has a wavelength much longer than the interatomic distance, will displace many electrons from their equilibrium position. However, the induced change in the magnetic flux swept by the electronic cloud is a tiny fraction of the electron areal density. The dimensionless $q_e$ is set by the product of these two numbers, one much larger than and the other much smaller than unity. The number of atoms perturbed by a transverse phonon with a wavelength of $\lambda_t$ is $\approx \lambda_t/a$. For each of these atoms, $\delta u_m/a$ represents the elongation of electronic bonds between atoms induced by the sound wave. Assuming that each atom shares one electron with its neighbor, this leads to

$$q_e \approx \frac{\delta u_m \lambda_{ph}}{a^2} \tag{21}$$

We saw above that $\delta u_m$ and $\lambda_{ph}$ both diverge at zero temperature. Therefore, this will also be true of their product and $q_e$. However, even at T=1 K, $\frac{\delta u_m(T)}{a} < 1$ and therefore $q_e$, remains a two digit number. Note that the decrease in the population of thermally excited phonons decreases as $\propto T^3$ , much faster than the $T^{3/2}$ enhancement of $q_e$ At 1 K, phonon-phonon interference is rare and almost all phonons travel ballistically without meeting other phonons.

Table 2 lists the established values for the three solids required to quantify the three parameters. For cubic silicon and germanium, with a single transverse mode, it is straightforward to extract them from equations 19, 20 and 21. For orthorhombic black P, there are two anisotropic transverse modes. The table lists the values obtained from the lowest and the largest velocities.

Given the neglect of numerous prefactors of the order of unity in the picture drawn above, it is surprising that the measured angles (the last column in table 1) and the expected angles (the last column in table 2) are so close to each other.

As for the other insulators displaying the thermal Hall effect, All of them contain atoms such as O, Cl , S, with masses comparable to Si or to P. In all, $T_{max}$ and $a$ are comparable to

the three cases under our scrutiny. Therefore, the relevant length scales and the expected Hall angle remain in the same range.

## 7   Concluding remarks

We started from two experimental observation: i) The Hall angle peaks at the peak temperature of longitudinal thermal conductivity; ii) The amplitude of this peak thermal angle is similar in different insulators. This led us to focus on the implications of a finite thermal Hall angle in section 2. Its finite value implies flow of heat without entropy production. In section 3, we saw that Normal phonon-phonon scattering events do not produce entropy and their frequency is most prominent at the peak temperature of longitudinal thermal conductivity. Therefore, they are suspected to play a role. Section 4 recalled that due to field-induced corrections to the Born-Oppenheimer approximation, phonons acquire a geometric (Berry) phase in a magnetic field. The consequences of the Berry phase for Normal events were discussed in Section 5. Because of this phase, magnetic field becomes a 'ratchet' creating an imbalance between absorption and emission Normal events. Finally, in section 6, the rough magnitude of the expected signal was found to match what has been experimentally observed.

In [44], Li, Guo, Zhu and the present author proposed that electric dipolar waves traveling with phonons can generate a thermal Hall effect. While the narrative thread of the present paper is different, the core idea in both is based on the combination of an electronic cloud and a point-like nucleus. The present scenario, which invokes phonon-phonon interference as a consequence of a complex phase, can provide an account of the temperature dependence and the order of magnitude of the observed signal.

In his 'Notes subject to ongoing editing' Resta [41] (circa 2022) commented that "The general problem of the nuclear motion—both classical and quantum—in presence of an external magnetic field has been first solved in 1988 by Schmelcher, Cederbaum, and Meyer [35]. It is remarkable that such a fundamental problem was solved so late, and that even today the relevant literature is ignored by textbooks and little cited."

To this very pertinent judgment, one may add that it is also remarkable that their consequences for phonons have not given much thought. If the phase of the wavefunction matters for molecules, why should it be neglected for a sound wave in a crystal ? The absence becomes even more surprising, given that Brown [84] had explicitly found an additional phase factor between the zero-field and finite field translation operators for Bloch electrons in a uniform magnetic field, . There is no trace of the implications of this phase factor for waves in insulators.

The scenario put forward in this paper postulates a phonon Hall effect, not captured by the Boltzmann transport equation and beyond semiclassical treatment, which appears to be the source of the experimentally observed signal in non-magnetic insulators. Inspired by Mead's treatment of molecules [37], it appears appropriate to call this phenomenon the 'lattice Aharonov-Bohm effect'. A direct experimental test of the present scenario may be achieved by scrutinizing acoustic wavefronts of silicon [85, 86] in the presence of an applied magnetic field.

## 8   Acknowledgments

The author thanks Benoît Fauqué, Xiaokang Li, Arthur Marguerite and Zengwei Zhu for helpful discussions.

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
