# Peer review of "Phonon thermal Hall as a lattice Aharonov-Bohm effect"

_SciPost Physics_

## Round 2 · Referee Report · Anonymous (Referee 2) · 2025-5-9

Strengths
Weaknesses
There are two points whose validity are still questionnable from my point of view: - the first one is on expression (15), where the field-induced phase shift takes only the geometrical vector potential into account, whereas it has to be added to the effect of the electromagnetic vector potential on the nuclei, leading to strong compensation effects.
- the second is on the estimation of the effective charge q_e (equation 21), where even the effect of bond stretching seems not valid.
Report
First point: Regarding expression (16), the author has introduced clearly the need of introducing a "geometric" vector potential (giving rise to the Berry phase) acting on the nuclear degrees of freedom, to compensate erroneous conclusions that would be drawn from the Born approximation. In the simplest cases, it was shown that this geometric vector potential allows to recover the intuitive result that the field has no effect (or only those due to the orbital magnetic susceptibility of the electron cloud) on neutral molecules (see Yin and Mead, Theor Chim Acta (1992) 82:397-406). Along the same line, it has even been shown for simple cases that the Berry phase is only a consequence of the Born approximation itself, and does not survive for exact calculations (Min et al. PRL 113, 263004 (2014) ). Moreover, this vector potential, as also explained by the author, is arising from the electronic cloud but acting on the nuclei. Hence, when discussing the field effect on phonons, with both electronic and nuclear degrees of freedom, equation (15) which describes the effect of the geometric vector potential, should be supplemented, at the qualitative level, by the effect of the field on the nuclei (at relative position with respect to the center of mass: $\vec{R}_n$, and charge $+Z_n e$):
$\exp\left[ +i \left( \vec{k} - \frac{e}{2 \hbar} \left( \vec{B} \times \sum_l\vec{r}_l\right) \right) \cdot \vec{R}_s \right] \rightarrow \ \exp\left[ +i \left( \vec{k} - \frac{e}{2 \hbar} \left( \vec{B} \times \left( \sum_l \vec{r}_l - Z_n \sum_n \vec{R}_n \right) \right) \right) \cdot \vec{R}_s \right]$
It would make it clear that at zero order, on a neutral phonon, there is no effect of the magnetic field, contrary to what the simple Born-Oppenheimer approximation would suggest for bare nuclei. The author makes the remark that the phase in equation (15) appears as a ratio of two flux, which is fine indeed. However, when he states that only the electron cloud induces and a flux as opposed to the point-like nuclei, this is probably not correct, and the modified expression above even shows that with no deformation, the electronic and nuclear flux should compensate.
Naturally, as phonons are the solid-state analog of molecular vibrations, the situation can be more complicated, due to the induced bond elongation.
Second point: This bond elongation has been indeed pointed out by the author as responsible for the large effective charge estimated by expression (21), together with the extension over a wave-lengfth of the deformation implying a large number ($\lambda/a$) of atoms.
However, this ignores two points: - first of all the compensation between electronic and nuclear charges discussed in Point 1, which shows that only "dipolar like" contributions can be expected to the flux, corresponding to differences between the sums of electronic and nuclear positions. - second, the elongation $\delta u_m$ is spread over the wavelength, hence the order of magnitude of the relative bond elongation is rather $\delta u_m/\lambda$ than $\delta u_m/a$. This already divides by $\approx 10$ the estimations made in the table 2. Moreover, it should even be the difference between bond stretching from one atomic position to the next which matters, which might bring another reduction factor to $q_e$, if does not even average to zero over the wavelength.
Hence, in the present form, some of the key qualitative arguments of the paper do not seem entirely valid, and the estimation of the order of magnitude of the effect does not seem justified. I continue to think that there are important ideas in this work worth publishing, notably regarding the potential role of normal collisions and anharmonicity, but I cannot be convinced by the more quantitative arguments.
Requested changes
Revising equations 15 and 21
Recommendation
Ask for major revision
Let me first thank you again for your time. Below are my answers to your two points.
First point
The referee calls into question the accuracy of equation 15, writing “…equation (15) which describes the effect of the geometric vector potential, should be supplemented, at the qualitative level, by the effect of the field on the nuclei (at relative position with respect to the center of mass”. The referee then adds: “The author makes the remark that the phase in equation (15) appears as a ratio of two flux, which is fine indeed. However, when he states that only the electron cloud induces and a flux as opposed to the point-like nuclei, this is probably not correct, and the modified expression above even shows that with no deformation, the electronic and nuclear flux should compensate.”
Response
i. Equation 15 and its corrected version proposed by the referee are identical (within an accuracy of ~10^-5). Since almost all the mass is concentrated in the nuclei and the sum of R_n (nuclear positions with respect to the center of the mass) is zero.
ii. Equation 15 was derived by Schmelcher et al. in 1988 (reference 35). Its accuracy has not been contested. In reference 41 (date 2022), Resta wrote: “The general problem of the nuclear motion—both classical and quantum—in presence of an external magnetic field has been first solved in 1988 by Schmelcher, Cederbaum, and Meyer [reference 35]. It is remarkable that such a fundamental problem was solved so late, and that even today the relevant literature is ignored by textbooks and little cited."
iii. What was contested was the interpretation of its significance. The referee refers to Yin and Mead, Theor. Chim Acta (1992) 82:397-406), who wrote: “The Born-Oppenheimer treatment of molecular systems in external magnetic fields has been considered in some detail by Schmelcher, Cederbaum, and Meyer. They showed that the conventional treatment leads to equations of motion in which the external field acts on bare nuclei which are completely unshielded by the electrons. By rephasing the electronic wave function so that, at least at infinite separation, one achieves a consistent gauge, they obtained what they called the "screened Born-Oppenheimer approximation," in which this defect is removed. As we shall see, this screening of the nuclei by the electrons can also (and equivalently) be understood as a manifestation of the geometric vector potential.”
iv. Equation 15 depends on the choice of the gauge. If instead of symmetric gauge, one chooses another gauge, say the Landau gauge, the phase becomes different. Scmelcher et al. framed their result as a matter of gauge centering. In reference 35, they wrote: “We therefore conclude that for a molecule in the dissociation limit, the gauge-centering phases …are closely related to the separate center-of-mass motions of the atoms in the molecule.” It was Mead who discovered that a geometric phase (no matter the gauge), will survive when molecules translate, rotate or vibrate.
Second point
The second point of the referee is about the link between bond elongation and the change in the magnetic flux during a vibration. Two objections are raised.
The first is that the relevant flux is the difference between the sums of electronic and nuclear positions. I think I have already addressed this objection by responding to the first point.
The second objection is that the elongation is spread over the wavelength and therefore the relevant distortion is proportional to the ratio of elongation and wavelength and not the ratio of elongation and lattice parameter.
I am very grateful to the reviewer for this comment, and I confess that this is a valid point. The elongation is indeed spread over the wavelength. At a given instant, not all atomic bonds are elongated by \delta u_m. Most of them are much closer to their equilibrium position. Therefore, the instantaneous distortion is much smaller than delta u_m/a. However, consider what happens during a whole cycle of vibration. During the temporal period of vibration, each atom involved will find itself at the crest of the wave. Therefore, integrating over the whole cycle, a distortion proportional to delta u_m/a will happen. It is true that this point needs to be clarified in the next version of the manuscript.
Let me recall that the estimation of q_e, is the most hand-waving part of this paper. I have no doubt that numerical prefactors are missing in 21. However, the point is that displaced electric charge involved in a collective atomic vibration giving rise to a transverse phonon can be estimated in a back-of-the-envelope manner. How can the periodic trajectory of the atomic wavefunctions be sensitive to the presence of a uniform magnetic field? The answer involves three length scales: the wavelength, the maximum displacement and the distance between atoms. The argument should integrate the whole temporal window during which an atom leaves its equilibrium position.

Xi Dai on 2025-05-07 [id 5453]
The paper is very well written and insightful. I thoroughly enjoyed reading it. I have the following comments and suggestions to help further improve the manuscript:
1. The author discusses two possible origins of the phonon thermal Hall effect: the molecular Aharonov-Bohm (AB) effect and lattice anharmonicity. The former gives rise to a non-dissipative Hall viscosity, which mixes longitudinal and transverse acoustic phonon modes, rendering their eigenvalues complex. This mixing leads to an “anomalous heat current,” analogous to the anomalous velocity in the electronic anomalous Hall effect, ultimately resulting in a transverse thermal current. Since this effect persists even at very low temperatures, the thermal Hall angle is expected to saturate to a small but finite value as temperature approaches zero. However, in most recent experiments, such a “residual Hall angle” is not observed. This likely suggests that Hall viscosity arising from harmonic-level considerations is not the dominant mechanism, and that third- or even fourth-order anharmonicity plays a significant role. It would be beneficial if the author could further elaborate on the low-temperature behavior of thermal Hall angles in light of this.
2. The author argues that the temperature at which the thermal Hall angle peaks coincides with the peak in longitudinal thermal conductivity, supporting the idea that strong anharmonicity contributes significantly to the effect. I found this argument very intuitive. I suggest the author extend this discussion by addressing the difference in temperature dependences of longitudinal and transverse thermal conductivities as well. Notably, the thermal Hall conductivity appears to increase more rapidly than the longitudinal thermal conductivity before the peak, and then decreases more sharply after the peak. Could this behavior also be attributed to phonon anharmonicity?
3. When a magnetic field is applied, the phonon thermal current acquires a transverse component, giving rise to the thermal Hall effect. However, what is the expected field dependence of the longitudinal thermal conductivity? Might it also be significantly affected by the magnetic field? If lattice anharmonicity indeed plays a central role, how should we expect the longitudinal conductivity to behave as a function of field strength?
Author: Kamran Behnia on 2025-05-14 [id 5483]
(in reply to Xi Dai on 2025-05-07 [id 5453])Thank you very much, Xi, for your generous comments and helpful feedback.
Indeed, the experimentally measured Hall angle does not saturate but peaks at finite temperature. My understanding is that this is because below the peak temperature, phonon-phonon collisions rarefy in comparison with other collision mechanisms. In clean crystals, this other mechanism is boundary scattering. In my opinion, what leads to the disappearance of the Hall angle is the vanishing \tau_B/_tau_N (See figure 3c of the paper). In other words, the thermal Hall angle decraeses at very low temperatures, because phonons do not meet other phonons frequently enough across their quasi-ballistic trajectory. In dirtier crystals disorder plays a signifcant role in this temperature range.
You are right. I will follow your advice. In the present version, the link between the experimental Figure 1d and the cartoon in Figure 3c has not been explicitly discussed.
An experimental answer is not available for all cases. There is one case in which I have been involved with and there the answer to your question is positive. In strontium titanate (See Fig. 1d in PRL 124, 105901 (2020)), there is a small but finite magnetothermal resistivity, even and quadratic in magnetic field. I agree with you that a comprehensive to the puzzle should address the field-induced change in both diagonal and off-diagonal components of the thermal conductivity tensor.
Anonymous on 2025-05-02 [id 5435]
Thanks for your paper.
You mention the longitudinal thermal conductivity kappa_ii depends on the size of the sample, but does the transversal kappa_ij depend on the sample size too? You seem not to mention this in the paper. Thanks.
Author: Kamran Behnia on 2025-05-02 [id 5440]
(in reply to Anonymous Comment on 2025-05-02 [id 5435])The low-temperature longitudinal thermal conductivity in crystals of silicon, germanium and black Phosphorus depends on the sample size. The transverse thermal conductivity of these crystals has not been experimentally measured as a function of sample size (neither in our lab nor in any place I know of). So the short answer to your question is that we don't know.
Note that such a size dependence is restricted to very clean insulating samples in which acoustic phonons become ballistic at low temperature. This is not the case of cuprates, strontium titanate , ruthenium chloride, and all other quantum materials in which a finite thermal Hall conductivity has been observed.
Anonymous on 2025-05-03 [id 5442]
(in reply to Kamran Behnia on 2025-05-02 [id 5440])Thanks for your reply.
Your paper focuses on the value of the thermal hall angle which is defined as kappa_ii/kappa_ij. So does it mean we don't know whether the Hall angle depend on the sample size?
Anonymous on 2025-05-07 [id 5461]
(in reply to Anonymous Comment on 2025-05-03 [id 5442])The answer to your question calls for a summary.
i) We know that kappa_ii (of clean crystals), below its peak (see figure 1a) , is size-dependent. We don't know about the size dependence of kappa_ij in this temperature range.
ii) Above the peak, in the so-called intrinsic regime , kappa_ii is NOT size-dependent and there is no reason to expect a size dependence of kappa_ij.
Therefore, it is safe to assume that above the peak, the thermal Hall angle (kappa_ij/kappa_ii) is an intrinsic quantity with no size dependence. But below the peak, boundary scattering dominates and this is no more true.

---

## Round 2 · Referee Report · Anonymous (Referee 1) · 2025-6-1

Strengths
The author has refined some of the arguments in the revised version, especially on the value of the effective charge.
Weaknesses
- Unfortuantely, I am still not able to follow central arguments of the paper, see report.
- Most likely, some relatively simple calculations could have helped to check central arguments of the paper. Such calculations are missing.
Report
Eq. (18), a simple Taylor expansion of Eq. (17) is supposed to show how a phase difference induced by a magnetic field, delta Phi_B, affects a phonon-scattering process (note that the factor 2 in the exponent of the first term of (18) is a typo).
As far as I understand, the author treats delta Phi_B in (17) just as a constant, shifting a plane-wave in space cos(k2 r-w2 t-delta Phi_B). Such a trivial shift of a plane wave can be absorbed in a simple shift of the point in space and time when and where this second phonon was created. Thus it appears to me that it cannot describe Hall-like effects. It is possible, that the author has something else in mind, but it remains unclear to me.
Another example is the equation for the effective charge (21) of a phonon. Here, the argument seems to be that stretching a bond by, e.g., 1% gives rise to a net charge of 1% of an electron charge (or, equivalently, gives rise to 1% of difference of enclosed fluxes for electron and ion motion). It is not fully clear to me whether this is correct or not, but I would have liked to see an argument which I am able to follow.
In my opinion, the paper would have benefitted a lot from some simple calculations. For example, consider a ionic crystal made from a heavy positive charge and a light negative charge (corresponding, e.g., to the electrons on a bond). It seems to me, that one can perform a simple calculation of the phonons in the presence of a magnetic field. Already the linear phonon theory would most likely allow to identify quantities like the effective charge q_e.
I am aware that the author is not a theorist but at least for a reader like me, it would help to check the heuristic arguments.
In conclusion, I cannot recomment the publication of the paper in scipost as I am not able to follow some of the most central arguments of the paper.
Recommendation
Reject
I do not attribute an "effective charge" to phonons. The focus of the paper are covalent insulators, not ionic ones. To see how complex charge distribution in a covalent insulator can become look at Figure 1 here :
https://journals.aps.org/prb/abstract/10.1103/PhysRevB.47.9385
Here is a one-sentence summary of the main idea:
If atoms move, the phase of the atomic wavefunction (as expressed by equation 17) will change, because it is impossible to keep the same center of mass for nuclei [heavy, point-like] and for electrons [light, broadly distributed] .
What is obscure or wrong about this?
To put it in another words, in my picture a phonon is as neutral as a photon , but it can couple to magnetic field by acquiring a phase (and not electric charge).

---

## Round 2 · Author Response

Both reviewers call the arguments “handwaving”. I have no objection to this qualification. Let me try to make them more transparent and dissipate a few misunderstandings.
1. The origin of equation 16
Referee 1 correctly notices that equation 16 is the central idea of the paper. However, this correct statement is followed by “The formula seems to suggest that a transverse phonon carries the charge q_e”. This is not accurate.
Let us start with equation 15, which is the equation 6.2 of ref. 35 and the result of a rigorous calculation.
Schmelcher et al. found that in the presence of the magnetic field, the molecular wavefunction in the Born-Oppenheimer approximation has a phase prefactor, which depends on the coordinate of the center of mass of the molecule and the sum of electron coordinates with respect to this center of mass.
Remarkably this phase factor does not depend on the ratio of the electron-to-nucleus mass ratio. Moreover, this expression is gauge dependent. Fig. 4a, illustrates that such a gauge-dependent phase factor is present for Born-Oppenheimer wavefunctions of a single atom, a molecule or a chain of atoms. In each case, one needs to consider the shift of the center-of-mass and to sum over the position of all electrons with respect to it.
Now, what happens to this phase factor in the presence of a collective vibration of atoms in a crystal? My answer begins by noting that the phase factor of Equation 15 is the ratio of two areas. The first is the product of center-of-mass and the sum of electron coordinates and the second is the square of the magnetic length. This inspires the following interpretation of equation 15:
The ultimate source of this phase factor is the difference in the spatial distributions of the positive charge of the nuclei and the negative charge of the electron cloud. The magnetic flux is negligible for the point-like nuclei, but not for the electron cloud. In the absence of compensation, in the presence of the magnetic field the molecular wave-function becomes a complex number.
Let me recall that the magnetic flux, in contrast to the vector potential, is gauge-invariant. This interpretation of Equation 15 leads to equation 16. A transverse phonon displaces electrons and atoms. An area can be readily defined by the product of the phonon wavelength and the maximum atomic displacement. Since equation 15 includes a sum over all electrons of the molecule, one should consider the number of electrons involved. This is the reason for the presence of the dimensionless parameter q_e in equation 16. This is not the “charge carried by a phonon”.
2. The order of magnitude of q_e
Referee 1 asks: “Where should these gigantic charges come from in a neutral solid?”
Referee 2 writes: “However, the presentation of the order of magnitude of the effect is not satisfying yet, due mainly to the estimation of q_e.”
Let us begin by recalling a fact. The well-known expression for the wavevector of a thermally excited acoustic phonon is the q=k_BT/\hbar v_s. Therefore, the wavelength is proportional to the inverse of temperature and diverges at the zero-temperature limit. This has never been considered as a problem. In realistic experimental set-ups, samples remain larger than the relevant phonon wavelength. Even in temperatures as low as 1 mK, the phonon wavelength is only a few microns.
The fact that the wavelength of an acoustic phonon at cryogenic temperatures becomes as long as several hundreds of angstroms has an important implication, however. Many atoms (and therefore many electrons) are displaced from their equilibrium position. Is it a surprise then that q_e is not a small number? After all, each oscillation of an acoustic phonon is a collective vibration of many atoms.
But the elongation of electronic bonds between neighboring atoms is small compared to the size of these bonds. In our range of interest, the phonon wavelength exceeds by far the interatomic distance and the latter is much longer than the maximum atomic displacement. Since q_e is their product, it ends up being of the order of unity.
Note that the increase in q_e with cooling (\propto T{-3/2}) is overcompensated by a decrease (\propto T^3) in phonon population. Therefore, the overall kappa_xy (like kappa_xx) is expected to smoothly vanish at low temperature.
3. Do Normal phonon-phonon collisions produce entropy?
Referee 1 writes:
“Cubic non-linearities are the main source for equilibration (without relaxing the total momentum) and thus they do produce entropy.”
I do not disagree with this statement. But there is a textbook consensus that in absence of Umklapp collisions, phonon-phonon interaction would not generate any thermal resistivity. More generally, the wave equation does not produce entropy, but the diffusion equation or any hybrid equation of wave and diffusion equations do (See for example Li et al., J. Non-Equilib. Thermodyn. 28, 279–291 (2003)). My point is that any interference between phonons keeping them pure solutions of the wave equation cannot produce entropy.
4. The phonon phase and the Hall angle
Referee 1 is surprised by the direct comparison between the Hall angle and the estimation of the phonon phase (Equation 16).
The idea is quite simple. Let us admit that phonons acquire a phase in the presence of a magnetic field. In this case, the amplitude of a sound wave, which is a modulation of density and pressure of matter in space, becomes a complex (and not a real) number. In laboratory environment (a field of 10 T and a temperature of few K) this phase is small (a few milliradians). Interestingly, this is comparable to the phase of photons in a typical Kerr experiment).
Neglecting prefactors of the order of unity, this small phase shift between colliding phonons sets the amplitude of rotation expected by phonon-phonon interference (as described in detail in section 5).

---

## Round 2 · List of Changes

-
The abstract now explicitly formulates the idea that the ultimate source of a field-induced phonon phase is the fact that the magnetic flux is negligible for the point-like nuclei, but not for the electron cloud.
-
Figure 4 has been modified. -The main text has been amended: The discussion of the route from equation 15 to equation 16 has expanded (section 4) and there is a now discussion of the divergence of the relevant length scales in the zero-temperature limit (section 6) . -One reference on the order of magnitude of Kerr rotation for polarized photons has been added.

---

## Round 3 · Referee Report · Anonymous (Referee 2) · 2025-7-10

Strengths

  • This work proposes an original route to explain how insulators can show a thermal Hall effect, a very actual subject as many systems are discovered that show a significant (reproducible !) thermal Hall effect, with no explanation for the finding. -T he paper focuses on giving physical insights and orders of magnitude to understand where the effect would come from.
  • It is clearly written, with pertinent references, and honestly acknowledge what is the most fragile part of the argument.

Weaknesses

  • The "least straightforward" step regarding the estimation of the Berry phase (now discussed "globally", with no specific point on the estimation of $q_e$, is still not convincing: notably formula (16) of the paper seem not consistent, physically (arguments on bond distortion) and mathematically (temperature dependence) with an effect arising from anharmonicity, proposed as a central and original part of this work, compared to previous theoretical estimates of the effect of the geometrical Berry phase of phonons under field.

Report

In this new version, the author tries to clarify the orders of magnitude of the phase change induced by the presence of a magnetic field on anharmonic phonons. One objection to the previous version of the paper was that many effects compensate hence the effect on the phase should be much smaller than the upper bound (equation (16) in this version v3): $\delta\phi_B \approx q_e \frac{\lambda_{ph} \delta u_m}{l_B^2}$.

A first compensation was that only departures between the center of mass of electrons and nuclei contributes to equation (15). It is indeed useless to introduce the center of mass of the nuclei in that equation, provided $\vec{R_s}$ can be time dependent: notably for anharmonic phonons, one could expect that the center of mass of the nuclei can oscillate with respect to the equilibrium $\vec{R_s}$ ? But this compensation between both electrons and nuclei motion makes it also hard for the referees to understand why $q_e$ should be so large, even if tens of atoms are involved over a wavelength.

In any case, the author agrees that what matters is the elongation of the “atomic bonds”, and most likely even, the distortion of the elongation as only anharmonicity can produce an effect. From this point of view, I could not understand how the new equations 17-20 do support the origin of equation (16), and more importantly, how they would show that instead of a simple upper bound, equation (16) gives a good order of magnitude of the effect?

Indeed, again, speaking of bond elongation, I do not see why one should consider, either in space or time, the difference in position of an atom at the crest and through of the (phonon) wave? Bond elongation refers to the change of distance ($a$) between two neighboring atoms, hence a simple estimate from a derivation of the spatial waveform shows that the effect is of order $(k.a)\delta u_m$. And speaking rather of the distortion of the elongation (the effect is zero for harmonic vibration), one should compare the difference in elongation between different bonds, so it might rather be a second derivative hence of order $(k.a)^2 \delta u_m$. This could already decrease the upper-bound by two orders of magnitude. By contrast, this factor $(k.a)^2$ would also restore the more intuitive result that $\delta\phi_B$ increases on warming, whereas in the present expression, $\delta\phi_B$ gets larger and larger at low temperatures, even though anharmonicity is less and less effective (as shown by a suppressed thermal dilatation below 70K in simple materials): at low temperature, there are much fewer phonons, so a small average displacement (as shown in [88]), nevertheless, a “large amplitude” ($\delta u_m$) for each phonon, but little anharmonicity because atoms move as a whole in acoustic waves and there is little distortion from site to site.

The connection between this microscopic bond distortion, and the value of the macroscopic Grüneisen parameter invoked at the end of paragraph 5 is also not clear for an average reader: there is no doubt that anharmonicity and thermal dilatation are deeply connected, and the Grüneisen parameter hence shows that the temperature variation of the characteristic phonon frequency is also connected to anharmonicity. But how does it justify formula (16)?

Regarding points which should still be clarified, the parallel made between anharmonicity and uniform electron distribution exemplified on figures 5a and b is surprising: in the Born-Oppenheimer approximation, on can indeed imagine the potential seen by electrons from the position of the ions. However, when discussing the motion of nuclei, where electrons will adapt quickly to their new position, the static electron density does not tell much about the non-anharmonicity, which, for the purpose of this paper, is strongly dependent on the amplitude of nucleus displacement due to thermal phonons. Hence, I fear again that the new material presented by the author does not help to clarify the pending questions of the previous version.

Requested changes

  • the estimation of the field-induced phase change still seems too "ad-hoc", and contradicts the central role of anharmonicity.
  • the new amterial added in this version did not help to support the authors arguments for the central formula (16).
  • hence, the last part on the order of magnitude needs revision.

Recommendation

Ask for major revision

  • validity: ok
  • significance: good
  • originality: high
  • clarity: high
  • formatting: good
  • grammar: excellent

Author:  Kamran Behnia  on 2025-07-11  [id 5636]

(in reply to Report 1 on 2025-07-10)

In this third report, the referee writes: “I could not understand how the new equations 17-20 do support the origin of equation (16), and more importantly, how they would show that instead of a simple upper bound, equation (16) gives a good order of magnitude of the effect?”

There is a misunderstanding here. Equation (16) is indeed “a simple upper bound” to the amplitude of the thermal Hall angle. Only, with q_e~1, it would “give a good order of magnitude” of the experimentally measured signal.

Equations 17-20 show that the Born-Oppenheimer phase of atoms induced by magnetic field can yield a geometric phase to their collective vibrations. However, for this to happen, atomic displacements with opposite signs during compression and at the expansion do not cancel out. Since all known solids are anharmonic, a finite q_e is expected to be ubiquitous.

I then argued that since the Gruneisen parameter of many solids is of the order of unity, cases of q_e~1 should be widespread. The referee finds that the connection to Grüneisen parameter is not clear. Moreover, the referee does not find the non-trivial distribution of valence charge in silicon (Figure 5b) to provide evidence for anharmonicity .

The attached file presents available data on elastic constants of silicon, togrther with a simple discussion.At least in the case of silicon, the data establishes that anharmonicity is large enough to cause a large difference (as large as average atomic displacement) between the crest and the trough of a collective vibration.

As for temperature dependence of the Hall angle, my main point is expressed in Figure 4c. The peak temperature is where Normal phonon-phonon collisions are most frequent. Equation 16 represents a bound in the absence of Umklapp and boundary scattering.

In the second report, the referee’s main objection was the inaccuracy of equation15. Unfortunately, there is no clear statement here to see how convincing my rebuttal was.

Attachment:

Anharmonicity_and_third-order_elastic_constants_in_silicon.pdf

---

## Round 3 · Referee Report · Anonymous (Referee 3) · 2025-7-20

Report

The manuscript is very well written and presents a compelling investigation into the phonon thermal Hall effect in nonmagnetic band insulators. The Introduction begins with an overview of the experimental observations of the thermal Hall effect, supported by a comprehensive list of references. It then transitions into a discussion of prior theoretical work, including connections to phonon angular momentum, chirality, and their coupling with magnetic materials.

The motivation for the present study stems from recent experimental observations of phonon thermal Hall conductivity in nonmagnetic band insulators such as phosphorus, silicon, and germanium. These findings suggest that the phonon Hall effect is a more ubiquitous phenomenon than previously thought and challenge the conventional understanding that phonons—as charge-neutral excitations—do not couple to magnetic fields. Despite this, many materials exhibit a finite thermal Hall response even in the absence of local magnetic moments.

The manuscript revisits this fundamental question and examines the breakdown of the conventional Aharonov-Bohm (AB) approximation in magnetic fields, arguing that phonons cannot be treated purely as neutral objects. Prior theoretical studies have already pointed out corrections to the AB approximation (Refs. 35, 36), and more recently, Ref. [43] proposed that these corrections result in a phonon Berry curvature that contributes to a finite thermal Hall conductivity.

The central result of this paper is the identification of phonon–phonon scattering and role of anharmonicity responsible for the observed thermal Hall effect. The authors highlight a key empirical observation: the temperatures at which the transverse (κij) and longitudinal (κᵢᵢ) thermal conductivities peak (T_max) are similar, and both conductivities decrease rapidly on either side of the peak. Despite differences in T_max across various materials, the ratio κ_ij/κ_ii (i.e., the Hall angle) remains bounded, as shown in Fig. 1(b), which includes materials such as Ge, Si, and P.

The paper addresses two key questions:

  1. Why does similar T_max appear in k_ii and k_ij?
  2. Why is the Hall angle bounded by a certain length scale rather than the phonon mean free path?

Both questions are answered in a clear and systematic manner, mainly based on empirical observations, and the proposed framework is accessible to a broad audience.

The discussion of the 3-phonon scattering process is particularly illuminating. The phase shift described in Eq. (15) and its lattice analog in Eq. (16) play a key role in understanding the microscopic origin of the thermal Hall effect. However, the connection between Eq. (16) and the following equations (17) - (20) is unclear, and gives the impression that Eq. (16), the main claim, remains to be resolved.

Requested changes

  1. It would be helpful to more clearly explain the relationship between Eqs. (16) and (17-20), to guide readers through the progression of the argument. Since it is empirical (q_e), rather than a rigorous theoretical derivation, it may be appropriate to explicitly state it and suggest some future theoretical studies.

  2. Since Fig. 1(b) includes materials with magnetic local moments, I suggest the authors explicitly clarify — perhaps in the figure caption — that the current theoretical framework focuses solely on phonon–phonon scattering and does not include spin–phonon or magnetic contributions. These effects may be relevant in the low-temperature regime (e.g., near magnetic phase transitions) for some of the materials included in Fig. 1(b).

Minor Comments:

  1. Page 9: In the sentence beginning “In 1988, Schemlcher,... BO approximation...,” a verb is missing. Please add “investigated,” “studied,” or similar.

  2. Page 12: In the last line, “Figure (taken from Ref. [79]),” please include the figure number (e.g., Figure 5).

  3. Page 14: When citing Ref. [43]’s estimation of the thermal Hall conductivity (10⁻⁶), please specify the temperature at which this estimate applies.

Recommendation

Ask for minor revision

  • validity: good
  • significance: high
  • originality: high
  • clarity: high
  • formatting: good
  • grammar: excellent

Author:  Kamran Behnia  on 2025-07-25  [id 5678]

(in reply to Report 2 on 2025-07-20)

Many thanks for your time, your positive assessment and your constructive criticisms. I would like to quickly address your two main criticisms: 1. "The relationship between Eqs. (16) and (17-20)." This was also an issue raised by the other referee and I admit that this part of the paper was hastily wrritten. The attached file is an attempt in clarification. Indeed, q_e is an empirical dimensionless parameter quantifuing anharmonicity.

  1. Yes ! I totally agree that "spin–phonon or magnetic contributions" are not accounted for in the present paper and they should indeed be important in "materials with magnetic local moments".

Hopefully, both these defficiencies will be amended in the next version of the paper.

Attachment:

anharmonicity_and_Berry_phase_of_phonons.pdf

---

## Round 3 · Author Response

This is a much more extensive revision than the previous one and this owes much to the reviewers’ stimulating criticisms.

Thank you!

There is an entire new section (numbered 5 and entitled ‘Field-induced geometric phase of phonons’) in this version. It tracks how an atomic/molecular Born-Oppenheimer phase can give rise to a phonon geometric phase. By doing this, I have followed this recommendation by referee 1: “Consider a ionic crystal made from a heavy positive charge and a light negative charge (corresponding, e.g., to the electrons on a bond). It seems to me that one can perform a simple calculation of the phonons in the presence of a magnetic field. Already the linear phonon theory would most likely allow to identify quantities like the effective charge q_e.”
The new section 5 (and the new Figure 5a) try to perform such a task. I hope that it becomes clear that the dimensionless parameter q_e is not an “effective charge”, but a measure of anharmonicity akin to the Gruneisen parameter. I admit that the previous version of the paper was neither clear nor accurate in this regard.
Referee 2 points out that the position of the nuclei is absent in Equation 15. However, as I discussed in more detail in my reply to their report, this does not make this equation incorrect. Since the mass is concentrated in the nuclei, the sum of R_n (nuclear positions with respect to the center of the mass) is zero. The referee’s remark was very helpful to reformulate the origin of the field-induced phase. It arises because of the unavoidable mismatch between electronic and nuclear center of mass. Note also that Equation 21 of the previous version has been deleted in the present version.
Let me sum up the proposed scenario:
Step 1: In the presence of magnetic field, Born-Oppenheimer approximation (according to which the atomic wavefunction is the product of an electronic and a nuclear wavefunction) holds only if the wavefunction acquires a field-dependent phase. This is not a new idea. But it has been overlooked in condensed matter physics.
Step 2: In anharmonic solids, this will generate a geometric phase for phonons. This phase is proportional to the magnetic field, the phonon wavelength and the maximum atomic displacement. There is a dimensionless prefactor, q_e, which depends on microscopic details. There are reasons to believe that its order of magnitude is of the order of unity. But this is not a proven statement.
Step 3: Such a phase would allow constructive and destructive interference between phonon modes. Therefore, in a three-phonon process, the “pseudo-momentum conservation sum rule” breaks down.
Any quantitative account of the thermal Hall angle and its temperature dependence requires a knowledge of the temperature dependence of the phase space of phonon-phonon collisions and the relative rate of Normal, Umklapp and Boundary scattering events and this is beyond the scope of the present work. But assuming that the dimensionless parameter in question is not far from unity, the rough amplitude and the rough temperature dependence of the experimentally measured thermal Hall angle would find an account.
I hope this is sufficiently clear.

---

## Round 3 · List of Changes

-Section 5 is entirely new.
-Figure 5 is entirely new.
-Figure 4 has been modified.
-Typos through the text have been corrected.
-The bastract and sections 4, 6 and 7 have been rewritten to enhance clarity.
-The equation defining q_e (21 in the previous version) has been deleted.

---

## Editorial Decision

resubmitted